# Mapping the co-evolution of artificial intelligence, robotics, and the internet of things over 20 years (1998-2017)

**Katy Börner**[1], **Olga Scrivner**[1]*, **Leonard E. Cross**[1], **Michael Gallant**[1], **Shutian Ma**[2], **Adam S. Martin**[1], **Lisel Record**[1], **Haici Yang**[1], **Jonathan M. Dilger**[3]

**1** Luddy School of Informatics, Computing, and Engineering, Indiana University, Bloomington, IN, United States of America, **2** Nanjing University of Science and Technology, Nanjing, Jiangsu, China, **3** Naval Surface Warfare Center Crane Division, Crane, IN, United States of America

* obscrivn@indiana.edu

**Data Availability Statement:** Data and algorithm details are provided in the Supplementary Information (SI) Appendix; survey instruments, ISI data, extracted keywords and code are at https://

## Abstract

Understanding the emergence, co-evolution, and convergence of science and technology (S&T) areas offers competitive intelligence for researchers, managers, policy makers, and others. This paper presents new funding, publication, and scholarly network metrics and visualizations that were validated via expert surveys. The metrics and visualizations exemplify the emergence and convergence of three areas of strategic interest: artificial intelligence (AI), robotics, and internet of things (IoT) over the last 20 years (1998-2017). For 32,716 publications and 4,497 NSF awards, we identify their topical coverage (using the UCSD map of science), evolving co-author networks, and increasing convergence. The results support data-driven decision making when setting proper research and development (R&D) priorities; developing future S&T investment strategies; or performing effective research program assessment.

## Introduction and prior work

Advances in computational power, combined with the unprecedented volume and variety of data on science and technology (S&T) developments, create ideal conditions for the development and application of data mining and visualization approaches that reveal the structure and dynamics of research progress. A critical challenge for decision makers is determining how to spend limited resources most productively. To do so, one must have a basic understanding of where the most productive research is being done, who the key experts are, and how others are investing in research. The identification of topics that have recently emerged as increasingly important or that are converging to create new synergies in research can be particularly fertile areas for research and development (R&D).

Different 'emergence' and 'convergence' definitions, indicators, and metrics have been proposed in previous work in this area. We use the four-attribute model of what constitutes technological emergence [1] and assume that emergent topics should evidence term novelty, persistence, and accelerating growth, and typically show the formation of a research

github.com/cns-iu/AICoEvolution. The table of all unique journal names and their (sub)disciplines is freely available online (https://cns.iu.edu/2012-UCSDMap.html). Publication raw data were downloaded via Clarivate Analytics Web of Science (WoS) (https://clarivate.com/webofsciencegroup/solutions/web-of-science/), an electronic resource governed by license agreements which restrict use to the Indiana University community. The extracted data and instructions are available at https://github.com/cns-iu/AICoEvolution. The WoS data was loaded using CNS's generic parser available on GitHub: https://github.com/cns-iu/generic_parser. The entity relationship (ER) diagram and data dictionary can be found at http://iuni.iu.edu/resources/web-of-science. NSF funding data is publicly available via NSF portal - https://nsf.gov/awardsearch and can be downloaded in bulk from https://nsf.gov/awardsearch/download.jsp. Intermediate and final results generated in our data preprocessing, statistical analysis and visualization are available as part of Supporting information (SI).

**Funding:** The work was partially funded by the National Institutes of Health grants U01CA198934 and R01LM012832 (K.B.), National Science Foundation (NSF) grants 1839167 and 1713567 (K.B.), and a Cooperative Agreement Award No. N00164-18-1-1005 supported by NSWC Crane CRANBAA18-005 (K.B, O.S). There was no additional external funding received for this study. The funders had no role in study design, data collection and analysis, decision to publish, or preparation of the manuscript. Any opinions, findings, and conclusions or recommendations expressed in this material are those of the author(s) and do not necessarily reflect the views of the sponsors.

**Competing interests:** The funders had no role in study design, data collection and analysis, decision to publish, or preparation of the manuscript. NSWC Crane did not play a role in the study design, analysis, and decision to publish. The specific role of Jonathan Dilger is articulated in the 'author contributions' section. This does not alter our adherence to PLOS ONE policies on sharing data and materials. Other authors were not employed by the NSWC Crane.

community. Much prior work exists on how to measure emergence. Guo et al. [2] proposed a mixed model that combines different indicators to describe and predict key structural and dynamic features of emerging research areas. Three indicators are combined: 1) sudden increases in the frequency of specific words; 2) the number and speed with which new authors are attracted to an emerging research area; and 3) changes in the interdisciplinarity of references cited. Applying this mixed model to four emerging areas for means of validation results in interesting temporal correlations. First, new authors enter the research area, then paper references become interdisciplinary, and then word bursts occur.

Recent work—including that funded by the US Intelligence Advanced Research Projects Activity (IARPA) Program on Foresight and Understanding from Scientific Exposition (FUSE) [3]—focuses on advanced linguistic techniques for identifying emerging research topics. Contributions from [4] include methods to extract terms from paper titles and abstracts and filter them based on 1) novelty, 2) persistence, 3) a research community, and 4) rapid growth in research activity. Porter et al. [5] developed emergence indicators that help 1) identify "hot topic" terms, 2) generate secondary indicators that reflect especially active frontiers in a target R&D domain, 3) flag papers or patents rich in emergent technology content, and 4) score research fields on relative degree of emergence. Other recent studies introduced several novel NLP methods to measure research diversity and interdisciplinarity. For example, topic modeling has been used to measure the degree of topic diversity [6], Shannon's entropy measure was applied to compute technological diversity using EU-funded nanotechnology projects data [7]. This paper uses robust and widely used topic identification methods and focuses on visualizing the emergence, co-evolution, and convergence of science and technology areas.

Convergence research was identified by the National Science Foundation (NSF) as one of the 10 Big Ideas for Future NSF Investments [8] that will help advance US prosperity, security, health, and well-being. In this paper, we present a repeatable procedure to characterize emerging R&D topics based on publication and funding data. Using this procedure, we visualize and analyze the convergence of three emerging R&D areas. Our efforts here both build upon and expand work by [9], who used the convergence to study scholarly networks for domains relevant for understanding the human dimensions of global environmental change.

A literature review and stakeholder-needs analysis were used to identify three domains of strategic interest: artificial intelligence (AI), robotics, and the internet of things (IoT). These three areas are of paramount importance not only for global prosperity, but also for defense and security [10]. AI, IoT, and robotics were named top technologies in 2018 with strong arguments and examples of how these technologies will drive digital innovation and completely transform business models [11, 12]. Since AI and robotics will have a major impact towards the future of economy, businesses need advanced preparation to meet these transformational challenges. The 2019 AI annual report pointed to the complexity of the fast-growing AI labor market: "unconditional convergence" and "unconditional divergence" in job demands at the same time [13]. In line with these developments, the White House prioritized funding for fundamental AI research and computing infrastructure, machine learning, and autonomous systems. Also, it argued for the need to work with international allies to recognize the potential benefits of AI and to promote AI R&D [14].

The paper is organized as follows. The following section details the stakeholder needs analysis, which guided the selection of strategic research areas and provided information about stakeholder insight needs. After that, we indicate both the datasets used and the data preprocessing needed for the study. This is followed by an outline of the methods used in the study and the results achieved. After examining validation design and insights in the user study, we end the paper with a discussion of the results and outlook for future research in this area.

## Stakeholder needs analysis

A stakeholder needs analysis (SNA) was employed in order to identify insight needs and use cases for using indicators and visualizations of emergence (and indirectly convergence) in daily decision-making environments. This study was approved by the Indiana University Institutional Review Board (protocol number 1807496060). Specifically, the SNA was designed to identify stakeholder demographics, task types, insight needs, work contexts, and priorities to better understand how decision makers might utilize static and dynamic information visualizations, topics of concern, and metrics currently used when making decisions. Twelve decision makers from the Naval Surface Warfare Center, Crane Division, (NSWC Crane) in Martin County, Indiana completed the one-hour survey. Participants included personnel from human resources, corporate operations, engagement, R&D, and various technical specializations. Surveys were conducted both on the Indiana University–Bloomington campus and at WestGate Academy, a technology park located adjacent to the naval base.

## Areas of strategic interest

In order to understand what topical areas were of interest, survey respondents were asked to identify which topics, from a list of eight, were most relevant for their work. Additional topics were also solicited from survey participants. The top eight topics are represented in Table 1. Each of the eight topics was queried via NSF award and Web of Science portals to identify the total of funding award and publications between 1998 and 2017.

Since we used funding and publication data to characterize emerging areas, the number of NSF awards and Web of Science (WOS) publications for each of the top-ranked areas was a factor in final topic selection (see Table 1). Note that for several areas, few funding awards have been made, allowing a human analyst to read through the related abstracts within a week. Artificial intelligence, the internet of things, and robotics are three domains identified as being of national strategic importance that have sufficient numbers of funding awards and publications for rigorous analysis.

Current metrics informing resource allocation decisions at NSWC Crane focus on internal R&D needs, recent calls for funding, and how other federal agencies are focusing their funding. Most respondents identified similar processes for allocating funding, deciding when to bring in outside expertise, and selecting contractors. This unified decision-making approach suggests that visualizations providing greater detail on how others are focusing their funding in strategic areas of interest have an important role to play in the process.

**Table 1. Topical interest and relevant funding and publication data.**

| Topic | #Experts that expressed interest | #NSF Funding awards active in 1998-2017 | #WOS Publications published in 1998-2017 |
|---|---|---|---|
| Advanced electronics | 10 | 122 | 206 |
| **Artificial intelligence** | 10 | 1,075 | 7,414 |
| Sensors and sensor fusion | 9 | 145 | 14 |
| **Internet of things** | 4 | 348 | 11,371 |
| Human systems integration | 4 | 0 | 102 |
| **Robotics** | 3 | 3,074 | 13,931 |
| System of systems test and evaluation | 3 | 4 | 0 |
| Power and energy management | 2 | 0 | 48 |

Users study report is available at https://github.com/cns-iu/AiCoEvolution.

In the second part of the SNA, participants were asked to view three sample visualizations, identify any insights gained from them, and elaborate on how they might use these insights. At the end, participants were asked to identify which of the three figures they found most useful and why. Seven respondents found the network visualization most useful, two found the tree map that identified top funders most useful, and one preferred a visualization showing the number of citations related to one topic over time. One participant responded that all were equally useful in different ways but were each not as useful on their own.

Participants were also presented with three interactive visualizations that they were able to manipulate on laptop computers or tablets. Again, participants were asked to identify insights gained from each visualization, and to elaborate on how they might use these insights. They were also asked to identify which of the three figures they found most useful and why. Six respondents found the co-authorship geospatial visualization most useful, three found the co-authorship network most useful, and two identified the science map as most useful. In sum, co-authorship networks in combination with geographic representations proved most interesting to the stakeholders surveyed. Additionally, stakeholder interest in interactive capabilities suggests that this would be a valuable direction for future efforts.

## Data and processing

A majority of publications related to each of the three target domains is captured in the Web of Science (WOS). In the US, much of the funding for these three focus areas is awarded by the National Science Foundation (NSF). The authors acknowledge that some subset of the current research in these areas is conducted through defense organizations and therefore can be difficult to capture in publicly available datasets. Only WOS publications and NSF awards from the last twenty years (1998-2017) were included in the analysis.

### Publications

Publication data was retrieved from the Clarivate Analytics Web of Science (WOS Core Collection) web portal and WOS XML raw data (Web of Knowledge version 5) acquired from Clarivate Analytics by the IUNI Science of Science Hub and shared through a Data Custodian user agreement with the Cyberinfrastructure for Network Science Center (CNS) on July 7, 2018. The total number of WOS publications is 69 million and there are more than one billion citation links from 1900 through the early months of 2018. Most publications have title, abstract, and keyword information that can be used for text mining. Publications also have a publication year and author data. Using the AuthorKeywords field, Clarivate Analytics extracted publication identifiers (accession numbers—UT) on October 24, 2019 containing the (compound) query terms "artificial intelligence," "internet of things," "IoT," and "robotics." We also manually evaluated the 371 keywords where the "IoT" term was semantically and structurally ambiguous (e.g., "interocular transfer (iot)," "antibiotic activity"), resulting in 64 false positives that were removed from the records. Clarivate UT identifiers were then split in eight segments and queried in the WOS web portal on January 24, 2019 to extract the ISI raw data that was then filtered by three keywords. Furthermore, two records have been removed in which the publication year was changed from 2017 to 2018: WOS:000425355100004 (artificial intelligence) and WOS:000425355100017 (IoT). The final number of records can be seen in Table 1. Two of the 7,414 papers (artificial intelligence) had no authors (WOS:000384456000001, WOS:000173337900025) and were excluded from the co-author and geospatial analyses. The number of publications per year and the number of citations for 1998-2017 are shown in Fig 1A. Note that the number of all publications (dashed line) are steadily increasing during the years 1998-2017, and between 2007-2011 we observe a nascent field of

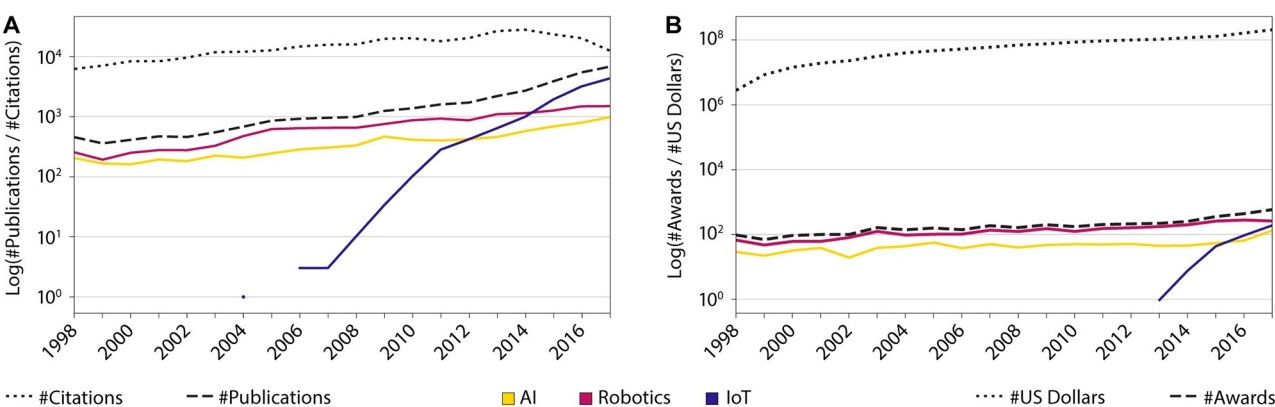

**Fig 1. WOS publications and NSF awards.** (A) Number of WOS papers extracted and the number of citations. Yellow solid line represents AI publications; red solid line, robotics publications; and blue solid line, IoT publications. The dotted line represents the total number of citations (total for all three domains). (B) Number of NSF grants and amount of funding awarded by NSF each year. Yellow solid line represents AI awards; red solid line, robotics awards; and blue solid line, IoT awards. The dotted line represents the total amount of funding (total for all three domains).

IoT (blue solid line) showing a sharp increase in the publication counts, exceeding AI publications (yellow solid line) by 2012 and robotics publications (red solid line) by 2014. The number of citations for all three domains (dotted line) shows a slow increase with a drop by the years 2015-2016. Note that papers published in recent years did not yet have time to acquire a high citation count.

There are many instances where publication records contain keywords from more than one focus area. For the terms "AI" and "robotics," for instance, there are 209 overlapping publication records. Between "AI" and "IoT" there are 46 overlapping publications, and between "IoT" and "robotics" there are 38. There were two publications containing keywords from all three areas ("A Novel Method for Implementing Artificial Intelligence, Cloud and Internet of Things in Robots" and "IT as a Driver for New Business"). Over time, there was a statistically significant increase in publications ($p < 0.0001$), totaling 32,716 during the period of 1998-2017 (see Fig 1A). Table 2 displays the number of unique keywords and authors for the three areas together with the totals.

## Funding

The NSF funds research and education in science and engineering through grants, contracts, and cooperative agreements to more than 2,000 colleges, universities, and institutions across the United States. It provides about 20 percent of the federal support academic institutions receive for basic research. More than 500,000 awards—including active, expired, and historical awards from 1952 to today—are available via the NSF award search portal.

NSF funding data for awards containing the (compound) terms "artificial intelligence," "internet of things," "IoT," and "robotics" was downloaded on July 24, 2018. Subsequently, the

**Table 2. Topical interest and relevant funding and publication data.**

| Topic | #NSF Unique Investigators | #NSF Unique Keywords | #WOS Unique Authors | #WOS Unique Keywords |
|---|---|---|---|---|
| **Artificial Intelligence** | 1,297 | 3,081 | 17,316 | 17,534 |
| **Internet of Things** | 575 | 2,435 | 23,691 | 21,204 |
| **Robotics** | 3,275 | 6,144 | 30,784 | 23,561 |

resulting sets were narrowed to NSF awards that were active in the last 20 years (Jan 1, 1998 to Dec 31, 2017): 1,075 in AI, 3,074 in robotics, and 348 in IoT (see Fig 1B and also Table 1). Note that 325 active awards have their start date earlier than 1998. Table 1 exhibits the number of NSF funding awards active in 1998-2017, and Table 2 illustrates the number of unique investigators and unique keywords for the three areas together with the totals.

There are some instances where awards overlapped: AI and robotics, for instance, received 146 funding awards, robotics and IoT received 17, while AI and IoT received only 2. There was no award for a project involving all three focus areas. The annual distribution demonstrates a statistically significant increase in awards ($p < 0.0001$), which seems to follow the same pattern as publications, lagging only by 10-fold on the log scale and totaling 4,497 awards over time (see Fig 1B). Note that the total award amount for the 1,075 AI funding awards is $494,310,951, the 3,074 robotics awards is $1,375,299,908, and the 348 IoT awards is $149,498,845. The number of awards and the amount of funding demonstrates a slow increase over time, with the exception of IoT showing a spike between the years 2013 and 2015.

### Keyword extraction via MaxMatch

Using results from a linguistic algorithm comparison detailed in [15], the MaxMatch algorithm was used to identify terms in NSF funding awards that match the unique WOS Author Keywords specific to the three topic areas. The MaxMatch algorithm [16] performs word segmentation to improve precision. The algorithm first computes the maximum number of words in the lexical resource (here NSF award titles and abstracts); then it matches long terms before matching shorter terms. Thus, given the text "artificial intelligence" and "intelligence" in a set of relevant terms, and "artificial intelligence" in the title and/or abstract of an award, the algorithm returns "artificial intelligence." This reduces oversampling of popular, short terms.

All keywords were pre-processed and normalized via Key Collision Fingerprint and ngram methods using OpenRefine [17]. This algorithm finds "alternative representations of the same things," thus allowing for the normalization of keywords (e.g., "internet of things (iot)," "iot— internet of things," "internet of thing"). We identified 1,739 clusters for AI; 2,333 clusters for IoT, and 3,201 clusters for robotics, which we then normalized by merging similar terms.

As a result, for WOS publications, we identified 55,946 unique *Author keywords*: 17,534 unique keywords for AI; 21,204 for IoT, and 23,561 for robotics. There are 2,935 terms that are shared between AI and robotics; 2,204 between AI and IoT; 2,331 between IoT and robotics; 1,117 shared across all three sets ($\sim$2% of all identified WOS keywords).

For NSF awards, we first excluded 325 active records where the start date was earlier than 1998. We then identified 9,185 unique *Author keywords* terms: 3,081 unique terms for AI; 2,435 for IoT, and 6,144 for robotics. Note that the keywords intersect: There are 1,376 terms that are shared between AI and robotics; 717 between AI and IoT; 914 between IoT and robotics, and 532 are shared between all three terms ($\sim$6% of all identified NSF keywords).

## Methods

The research was conducted under IRB protocol 1807496060 and protocol 1809442778.

### Burst detection and visualization

A Burst detection algorithm helps identify sudden increases in how often certain keywords are used in temporal data streams [18]. The Kleinberg's algorithm, available in the Sci2 Tool [19], reads a stream of events (e.g., time-stamped text from titles and abstracts) as input. It outputs a table with burst beginning and end dates and a burst weight, indicating the level of 'burstiness.'

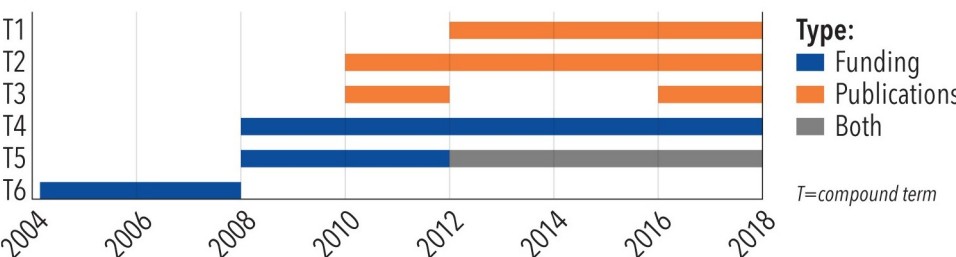

**Fig 2. Burst visualization using horizontal bar graphs.**

Burst weights can be used to set thresholds (for example, to keep only the top-10 words with the highest burst weights). For this study, burst was run with gamma at 1.0, density scaling at 2.0, and bursting states and burst length of 1. The weights of terms that burst multiple times were summed up before top-n were picked for visual graphing. Note, however, that the original burst values are used to code the bars by area.

A novel burst visualization was implemented to show the temporal sequence of bursts in funding and publication data as well as co-bursts in both types of data. Using the temporal bar graph visualization as a starting point, each bursting term is represented as a horizontal bar, where the length represents burst duration (defined by a specific start and end date), height (thickness) shows burst strength, and color denotes data type (e.g., funding, publications). Fig 2 explains this new burst visualization using a hypothetical example of six compound terms (T) that burst between 2004-2018. The area of each bar encodes the burst weight equally distributed over all years in which the burst occurs. Note that some burst terms are consecutive—they end in year x and start in year x+1—and might have different burst weights. Bars are color-coded by type with blue indicating bursts in funding, orange bursts in publications, and gray denoting simultaneous bursts in both types of data. While two plots side-by-side would make each easier to examine, this combined visualization makes it possible to compare bursts (start and end dates, burst weights, co-bursts) across datasets.

## Top organizations and funding

The Web of Science (WOS) online portal [20] supports searches for specific topics and facilitates the retrieval and examination of top funding agencies and top research organizations. Between September 11 and October 23, 2018, queries were run on the terms "artificial intelligence," "internet of things," "IoT," and "robotics" using topic and title fields for the years 1998-2017. The 'Organization-Enhanced' field was used which returns all records including name variants and preferred organization names. Organizations and funding agencies come with a variety of spellings. For example, the NSF and National Science Foundation show up as two different entities making name unification necessary. For each of the three focus areas, we selected the top-10 research institutions and funding agencies, identified their country, and tabulated results (see S1 Table in S1 Appendix).

## Co-author networks

The Sci2 Tool was used to extract a co-author network using the co-author column. The resulting, undirected, weighted network has one node type: authors. Each node represents a unique author. Edges represent co-occurrence of authors on a paper (i.e., the relationship between pairs of authors to be co-authors or not). Edge weights denote the number of times two authors co-occur on (i.e., co-author) a paper. The example given in Fig 3 illustrates a table

| Paper | Authors | Year |
|-------|---------|------|
| P1 | A1 | 1970 |
| P2 | A2;A6 | 1980 |
| P3 | A1;A3 | 1990 |
| P4 | A1;A4;A5 | 1995 |
| P5 | A2;A5;A6 | 1995 |

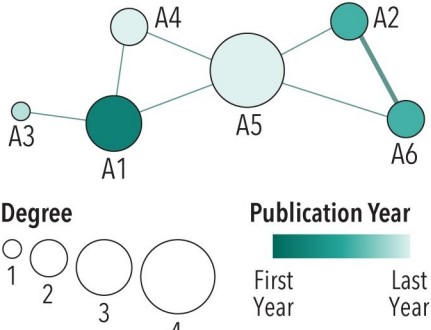

**Fig 3.** Co-authors listed on papers (left) are rendered as a co-author network (right).

with five papers by a total of six authors. Authors A2 and A6, for example, co-occur on papers P2 and P6, so their link is twice as thick. All other edges have a weight of one. For this study, we used perfect match on names in the *Authors* field using the 'Extract Co-Occurrence Network' functionality.

Force directed network layout takes the dimensions of the available display area and a network as input. Using information on the similarity relationships among nodes—for example, co-author link weights—it calculates node positions so that similar nodes are in close spatial proximity. Layout calculations are computationally expensive as all node pairs have to be examined and layout optimization is performed iteratively. Running the Generalized Expectation Maximization (GEM) layout available in Sci2 via GUESS on a simple network of six authors that published five papers (Fig 3, left) results in the layout shown on the right in Fig 3. The network has six co-author nodes that are fully connected. Author nodes are size-coded by the number of co-authors and color-coded by the year of the first publication. Edges denote co-authorship relations and are color coded by year of the first joint publication and thickness coded by the number of joint papers. The legend communicates the mapping of data variables to graphic variables.

WOS publications provide affiliation data (addresses) for authors, making it feasible to generate network overlays on geospatial maps (see Fig 4). We use Make-a-Vis [21, 22] to generate latitude and longitude values for author addresses as well as co-author networks; authors with no US address are excluded from this network. If an author has multiple addresses, the most recent address is used. The co-author network with geospatial node positions is saved out and read into Gephi [23] for visualization using Mercator projection; with node area size indicating number of citations, node color indicating the first year published, and edge thickness representing the number of times two authors are listed on a paper together.

## Science map and classification system

The UCSD Map of Science and Classification System was created using 2006–2008 data from Scopus and 2005–2010 data from the Web of Science [24]. The map organizes more than 25,000 journals/conference venues into 554 (sub)disciplines that are further aggregated into thirteen main scientific disciplines (e.g., physics, biology), which are labeled and color-coded in the map. For example, the 'Math & Physics' discipline in the top left has a purple label and all subdiscipline circles are rendered in purple. The network of 554 (sub)disciplines and their major similarity linkages was laid out on the surface of a sphere and flattened using a Mercator projection, resulting in a two-dimensional map (Fig 5). The UCSD Map of Science wraps around horizontally (i.e., the right side connects to the left side of the map).

| Authors | City | Lat/Lon |
|---------|------|---------|
| A1 | Los Angeles (CA) | 34/-118 |
| A2 | Austin (Texas) | 30/-97 |
| A3 | Santa Clara (CA) | 37/-121 |
| A4 | Ft Lauderdale (FL) | 26/-80 |

| Papers | Authors | #Citations | Year |
|--------|---------|-----------|------|
| P1 | A1;A2 | 6 | 2000 |
| P2 | A1 | 300 | 2005 |
| P3 | A1;A3,A4 | 20 | 2010 |

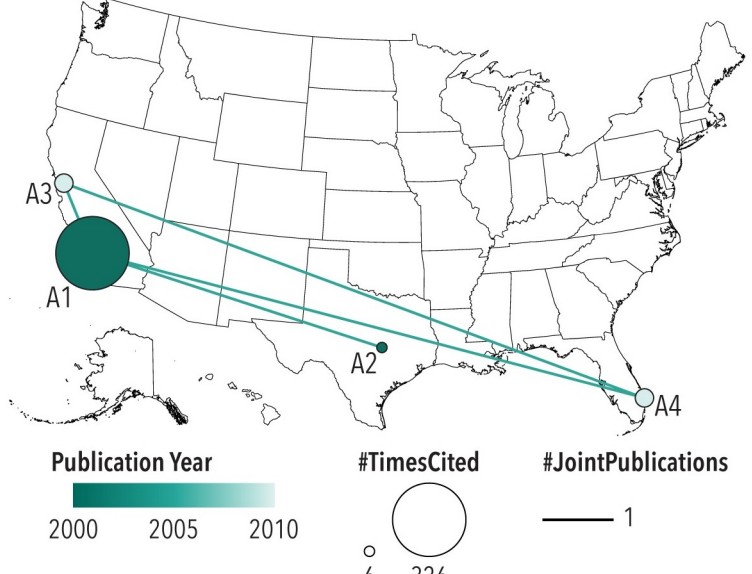

**Fig 4. Co-author network overlay on U.S. geospatial map.** See GitHub Appendix for more information on the co-author and geospatial network workflow https://github.com/cns-iu/AICoEvolution. Note: US-Atlas is under an ISC license. Copyright 2013-2019 by Michael Bostock.

In order to create proportional symbol data overlays, a new dataset is "science-coded" using the journals (or keywords) associated with each of the 554 (sub)disciplines. For example, a paper published in the journal *Pharmacogenomics* has the science-location 'Molecular Medicine,' as the journal is associated with this (sub)discipline of the discipline 'Health Professionals.' If non-journal data (e.g., patents, grants, or job advertisements) need to be science-located, then the keywords associated with each (sub)discipline can be used to identify the science location for each record based on textual similarity. In this study, multidisciplinary journals such as *Science* or *Nature*, which are fractionally assigned to multiple disciplines, were associated with a 'Multidisciplinary' discipline. Journals that cannot be classified are automatically associated with an 'Unclassified' discipline. Bar graph visualizations showing the number of papers and citations for the 15 disciplines are used to support comparisons (see S1-S3 Figs in S1 Appendix).

| Year | Journal Title |
|------|---------------|
| 2005 | J1 |
| 2006 | J2 |
| 2007 | J3, J4 |
| 2008 | J5 |
| 2009 | J6, J7 |

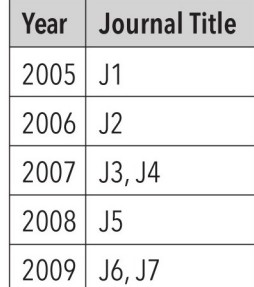

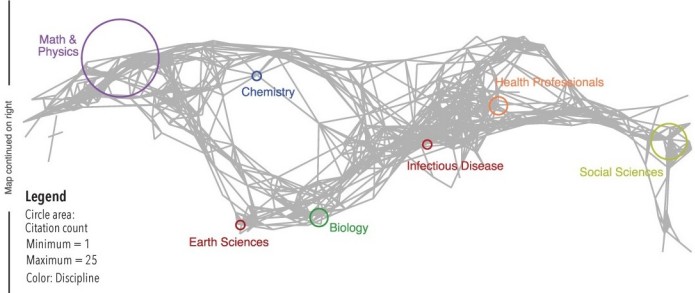

**Fig 5.** A process of mapping scientific journal names into a discipline and (sub)discipline topic (left) and the projection of journal topics into 2D spatial position (right).

## Results

The previous sections gave a general introduction to this research, prior work, stakeholder needs, available datasets, and methods used. This section represents results for each of the three areas separately, compares those results, and analyzes, visualizes, and discusses convergence of the three fields.

### Artificial intelligence (AI)

The field of artificial intelligence studies the interplay of computation and cognition. It is concerned with subjects such as knowledge representation and retrieval, decision-making, natural language processing, and human and animal cognition. AI research generates tools and artifacts to address problems involving complex computational models, real-world uncertainty, computational intractability, and large volumes of data. It also uses computational methods to better understand the foundations of natural intelligence and social behavior.

Top-funded AI awards include *BEACON: An NSF Center for the Study of Evolution in Action* led by Erik Goodman at Michigan State University, active 2010-2021, total amount awarded to date $43M; the *Center for Research in Cognitive Science* led by Aravind Joshi at the University of Pennsylvania, 1991-2002, $21M; and the *Spatial Intelligence and Learning Center (SILC)* led by Nora Newcombe at Temple University, 2011-2018, $18M.

**WOS-top organizations and funding.** The top-10 AI-funding organizations most frequently acknowledged in papers (seven were merged, see S6 Table in S1 Appendix) and the top-10 research organizations are exhibited in Fig 6 (top right). The leading top funders are the Natural Science Foundation of China and the National Science Foundation in the U.S., followed by agencies from the UK, Mexico, Europe and Brazil. The top research organizations are the French National Center of Research and the French University of Cote-d'Azure. U.S., India, China, and United Arab Emirates are among countries in the top-10 leading AI research institutions. The list of abbreviations for agencies and institutions and their name variations is exhibited in S6 Table in S1 Appendix.

**Burst of activity.** Within the 1,075 NSF awards that have the keyword "artificial intelligence," there are 161 bursts. There are no terms that burst twice. As for the 7,414 WOS publications with the keyword 'artificial intelligence,' there are 89 bursts total with no terms bursting twice. There are eight overlapping bursts for NSF and WOS keywords: "Agents," "Big Data," "Component," "Control," "Deep Learning," "Expert Systems," "Machine Learning," and "Psychology." The top bursting activity for NSF is "Machine Learning" with the burst weight of 13.04, while for WOS it is "Learning (Artificial Intelligence)" with the burst weight of 39.29. Among the top-10 bursting activities, "Big Data" co-bursts in both sets in 2014-2017 and is rendered in gray (see Fig 6, top left). The other two co-bursting terms are "Deep Learning" and "Machine Learning." Burst weight is indicated by bar thickness with "Learning (Artificial Intelligence)" having the strongest burst in 2015-2017. Interestingly, the bursting activity between 1998 and 2007 is predominately present in WOS publications (orange color). Starting with the keyword "Web," NSF awards exhibit bursting activities, culminating by co-bursting with publication from 2014 in "Big Data," "Machine Learning" and "Deep Learning."

**Key authors and collaboration networks.** The complete co-author network for AI has 17,316 unique author nodes. There are 437 authors with more than three papers, 235 authors with more than four papers, 143 with more than five papers. Of these nodes, 901 are not connected to any other node (they are called isolates) denoting that these 901 authors have not co-authored with any others during the 20 years. There are 31,476 co-author edges. The average degree is 3.64. The network has 4,299 weakly connected components, including the 901 isolates. The largest connected component consists of 1,914 nodes and is too large to visualize in

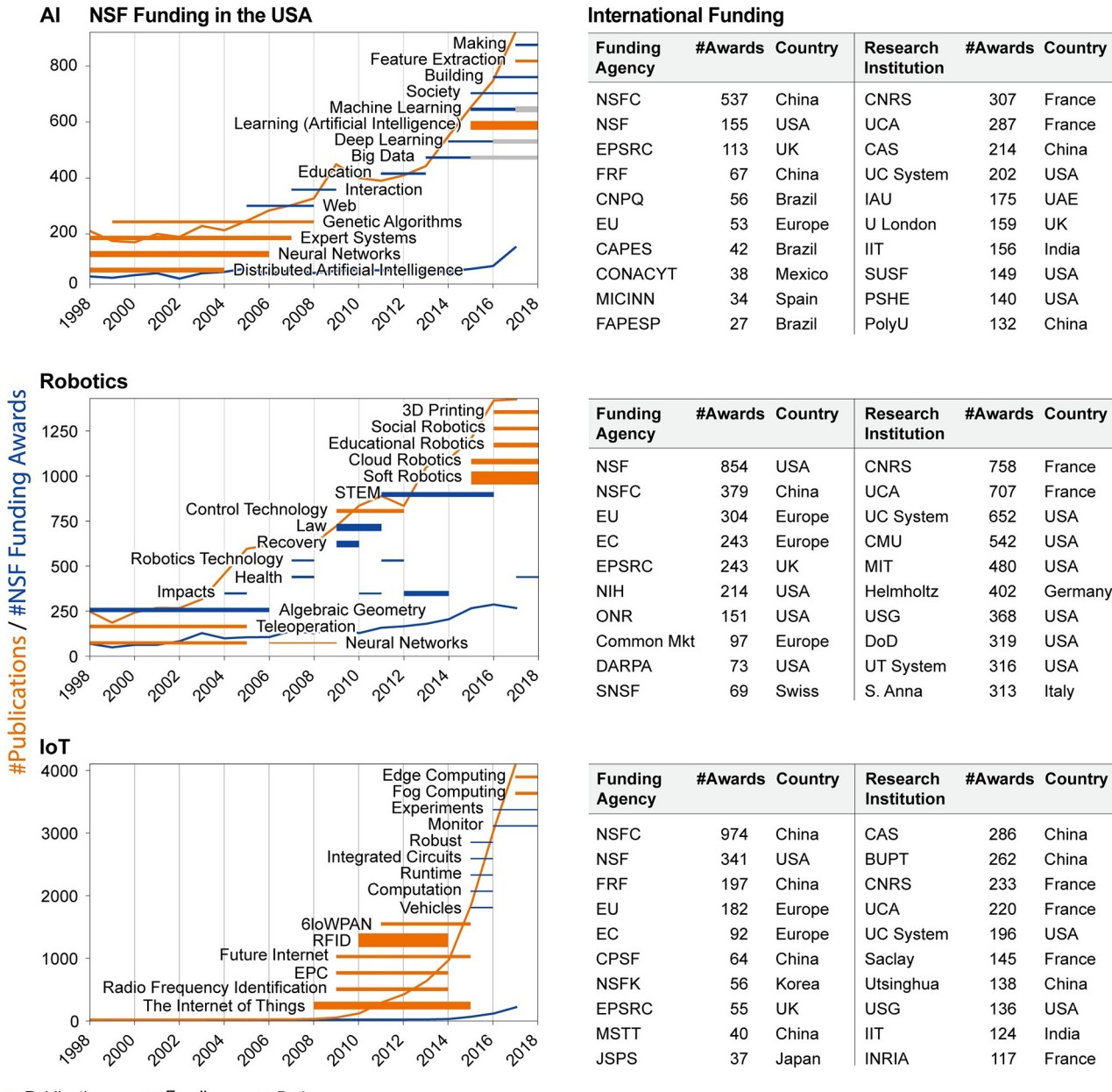

**Fig 6. Bursts of activity and top funding organizations.** Shown on the left are the top-15 keywords with the strongest bursts in funding awards (blue color) and publications (orange color). Gray color indicates a double burst. For example, "Machine Learning" is bursting in both publications and funding awards in 2017-2018. Bar thickness indicates the strength of each burst (weight). Given on the right are the Top-10 funders and research organizations associated with publications (see S5 Table in S1 Appendix).

a paper. The network was filtered by times-cited ≥ 1 resulting in 11,166 nodes, 21,280 edges, 2,763 weakly connected components and 473 isolates. The largest connected component of this network has 675 co-authors (with 473 isolates removed) and is shown in Fig 7 (top left). Author nodes and node labels are size-coded by the number of citations. Links, which denote co-authorship relations, are thickness-coded by the number of joint publications. The total number of links is 2,028 and they are filtered by the number of co-authored network (≥1).

**AI**

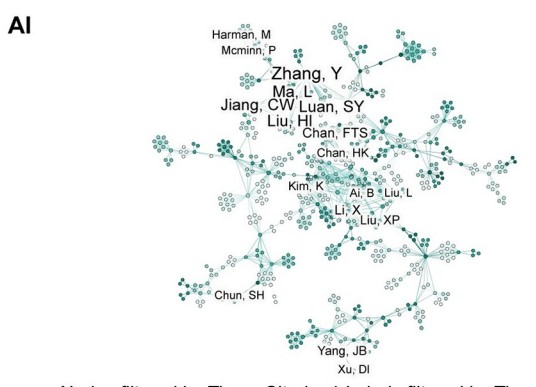

Nodes filtered by Times Cited ≥ 1 Labels filtered by Times Cited ≥ 100.
The largest selected component: 695

**Robotics**

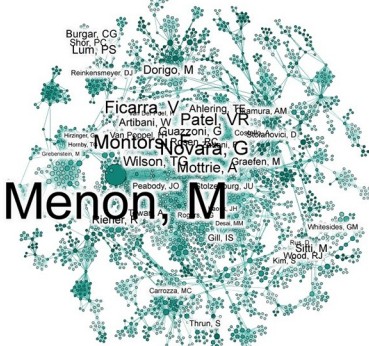

Nodes filtered by Times Cited ≥ 20 Labels filtered by Times Cited ≥ 850
The largest selected component: 3007

**IoT**

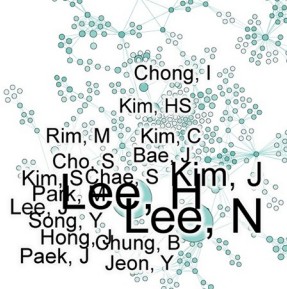

Nodes filtered by Times Cited ≥ 5 Labels filtered by Times Cited ≥ 400
The largest selected component: 585

**#TimesCited**

3,890    1,949    ○ 1

**Publication Year**

1998    2013    2017

### Top-10 US Authors by #Citations
*Before any filters were applied*

| Name | #Publications | #Citations | City | State |
| --- | --- | --- | --- | --- |
| Pennebaker, JW | 1 | 735 | Austin | TX |
| Coello, CAC | 9 | 537 | New Orleans | LA |
| **Zhang, Y** | 13 | 475 | Springfield | IL |
| Kisi, O | 8 | 427 | Butler | GA |
| Smith, SF | 1 | 393 | Cambridge | MA |
| Swaminatathan, JM | 1 | 393 | Berkeley | CA |
| Cook, DJ | 3 | 375 | Arlington | TX |
| Bridewell, W | 2 | 338 | Pittsburgh | PA |
| Chapman, WW | 1 | 328 | Pittsburgh | PA |
| Augusto, JC | 4 | 322 | Pullman | WA |

| Name | #Publications | #Citations | City | State |
| --- | --- | --- | --- | --- |
| **Menon, M** | 48 | 4,357 | Detroit | MI |
| Krebs, HI | 38 | 2,805 | Cambridge | MA |
| **Novara, G** | 14 | 2,302 | Orlando | FL |
| **Montorsi, F** | 13 | 2,301 | Hamburg | NY |
| **Patel, VR** | 32 | 2,177 | Birmingham | AL |
| **Mottrie, A** | 18 | 1,856 | Wilmington | DE |
| Hogan, N | 18 | 1,804 | White Plains | NY |
| **Wilson, TG** | 13 | 1,753 | Duarte | CA |
| **Lum, PS** | 18 | 1,623 | Irvine | CA |
| **Guazzoni, G** | 7 | 1,538 | Orlando | FL |

| Name | #Publications | #Citations | City | State |
| --- | --- | --- | --- | --- |
| Xu, LD | 38 | 3,651 | Norfolk | VA |
| Palaniswami, M | 6 | 2,651 | Houghton | MI |
| Chen, M | 14 | 1,098 | Flint | MI |
| He, W | 14 | 1,095 | Norfolk | VA |
| Guizani, M | 7 | 968 | Pittsburgh | PA |
| Al Fuqaha, A | 4 | 927 | Kalamazoo | MI |
| Mohammadi, M | 2 | 903 | Chicago | IL |
| Zaslavsky, A | 17 | 785 | Worthville | PA |
| Jara, AJ | 47 | 715 | Norcross | GA |
| Bobadilla, J | 1 | 693 | Cornelius | NC |

**Fig 7.** Co-author network and top-10 authors by #Citations for AI (top), robotics (middle), IoT (bottom).

The labels are filtered by the number of times-cited (≥100). Author 'Zhang, Y' has the largest number of citations (475) in this network. Zhang is also one of the top-10 cited US authors in the complete co-author network (see Fig 7 top table).

When presented with key author and collaboration networks during the stakeholder needs analysis, users indicated that network diagrams overlaid on top of geographic maps provided greater insights than network diagrams not anchored to geographic space. In response to that

**AI**

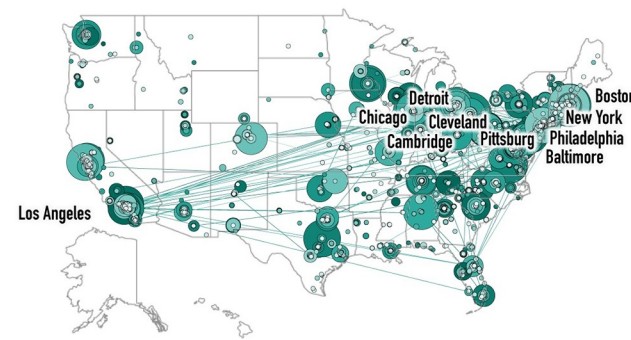

### Top-10 Cities by #Citations

| City | State | #Authors | #Publications | #Citations |
|------|-------|----------|---------------|------------|
| Austin | TX | 41 | 10 | 910 |
| Pittsuburgh | PA | 79 | 28 | 888 |
| Baltimore | MD | 35 | 13 | 483 |
| Chicago | IL | 48 | 15 | 483 |
| Berkeley | CA | 24 | 10 | 431 |
| Troy | NY | 20 | 10 | 392 |
| Tampa | FL | 24 | 8 | 364 |
| Santa Clara | CA | 11 | 5 | 338 |
| Denver | CO | 39 | 17 | 288 |
| San Francisco | CA | 25 | 12 | 285 |

Edges filtered by Joint Publication ≥ 1. Top 10 cities are aggregated by #Citations

**Robotics**

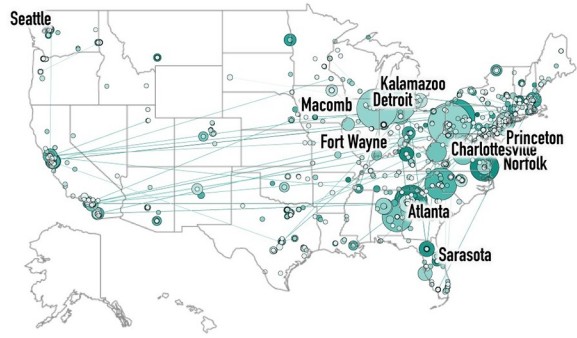

| City | State | #Authors | #Publications | #Citations |
|------|-------|----------|---------------|------------|
| Pittsburgh | PA | 777 | 196 | 6,486 |
| Cambridge | MA | 614 | 147 | 5,672 |
| Baltimore | MD | 459 | 86 | 3,767 |
| Los Angeles | CA | 477 | 97 | 3,121 |
| Detroit | MI | 331 | 52 | 2,482 |
| New York | NY | 482 | 114 | 2,407 |
| Cleveland | OH | 515 | 99 | 2,368 |
| Boston | MA | 333 | 73 | 2,254 |
| Philadelphia | PA | 281 | 77 | 2,235 |
| Chicago | IL | 368 | 70 | 1,841 |

Edges filtered by Joint Publication ≥ 5. Top 10 cities are aggregated by #Citations

**IoT**

| City | State | #Authors | #Publications | #Citations |
|------|-------|----------|---------------|------------|
| Kalamazoo | MI | 5 | 1 | 902 |
| Norfolk | VA | 17 | 8 | 635 |
| Atlanta | GA | 169 | 42 | 446 |
| Charlottesville | VA | 11 | 5 | 440 |
| Princeton | NJ | 22 | 7 | 268 |
| Sarasota | FL | 3 | 1 | 250 |
| Seattle | WA | 43 | 13 | 249 |
| Detroit | MI | 5 | 1 | 247 |
| Ft Wayne | IN | 12 | 2 | 188 |
| Macomb | IL | 3 | 2 | 178 |

Edges filtered by Joint Publication ≥ 3. Top 10 cities are aggregated by #Citations

**#TimesCited** 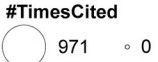

**Publication Year** 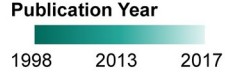

**Fig 8. Co-author network extracted from WOS publications and overlaid on the U.S. map.** Note: US-Atlas is under an ISC license. Copyright 2013-2019 by Michael Bostock.

feedback, we present here key authorship and collaboration network diagrams that use the United States as a base map.

**Co-author U.S. map.** To overlay the obtained co-authored network over a U.S. map, we used Make-a-Vis [21]. Fig 8 (top left) shows the number of co-authors for AI with nodes representing the number of citations and a darker hue showing the first year of publication for a given author. The network concentration is noticeable in the eastern states, and Austin and

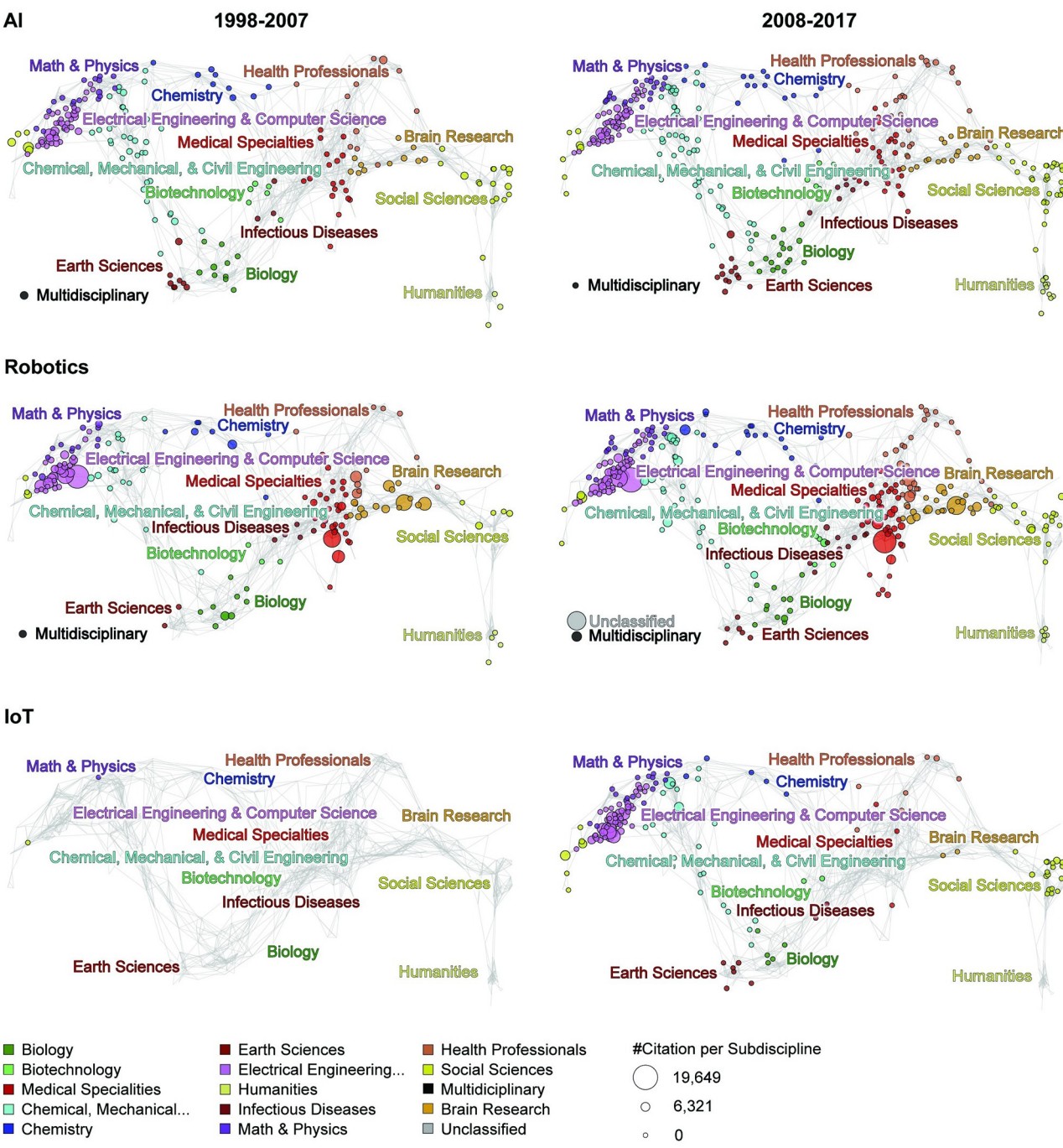

**Fig 9.** Topical coverage of AI, robotics, and IoT publications published in 1998-2007 (top) and 2008-2017 (bottom).

Pittsburgh are the top-two cities in the mid-US, with AI research cited 910 and 888 times, respectively (see Fig 8, top right).

**Topical evolution.** The topical distribution of 7,414 WOS publications on AI is shown for two 10-year time slices in Fig 9 (top) using Make-a-Vis. The UCSD Map of Science and Classification system used with identical circle area size coding (see discussion in the Methods

section). Most of the papers are in the 'Electrical Engineering & Computer Science' disciplines and in the 'Chemical, Mechanical, and Civil Engineering' disciplines. Note the increase of papers in the 'Social Sciences' and 'Health Professionals' (see also S1 Fig in S1 Appendix). The top-five most cited papers, along with their publication year and total number of citations, are shown in S4 Table in S1 Appendix.

Fig 9 illustrates the evolution of topical coverage for each of the key terms. The comparison between left (1998-2007) and right (2008-2017) panels indicates the evolution of each term within scientific disciplines. The topical coverage for AI has increased for all scientific disciplines with the highest publication change in 'Electrical Engineering & Computer Science' (10,391), 'Chemical, Mechanical & Civil Engineering' (6,283), and 'Social Sciences' (2680) (see also S1 Fig and S1 Table in S1 Appendix).

## Robotics

**WOS-top organizations and funding.** The top-10 funding organizations (seven were merged, see S6 Table in S1 Appendix) and the top-10 research organizations are given in Fig 6 (right panel). As can be seen, the U.S. is clearly providing the largest amount of funded publications, while the strongest research institutions are in France. The Chinese funding organization NSFC is the second top-funding agency; however, no Chinese research institution is listed among the top-10. The University of California System (UC System), the University System of Georgia (USG), and MIT are among the top-10 research institutions with the most papers.

**Burst of activity.** In the 3,074 NSF awards, there are 654 total bursts, with 28 double, and two triple bursts ("CPS" and "Impacts"). Summing up burst weights of double and triple bursts results in the top-15 bursts, rendered in blue in Fig 6. As for the 13,931 WOS publications, there are 261 bursts total with seven double bursts. The top-15 bursts are rendered in orange in Fig 6 (middle). Between NSF and WOS keywords, there are 47 overlapped keywords. However, there is no overlapping between the top-15 bursts.

Burst weight is indicated by the bar thickness, with "Soft Robotics" having the strongest burst in 2014-2017 for WOS with the weight of 39.3 and "Law" for NSF with the weight of 31.6. "STEM" is the longest most recent NSF burst between 2010 and 2015.

**Key authors and collaboration networks.** The original dataset has 30,784 unique authors. There are 2,363 authors with more than three papers, 1,531 authors with more than four papers, and 1,096 with more than five papers. There are 621 isolates, authors who have not co-authored with any others during the 20 years. There are 96,982 co-author edges. The average degree is 6.30. The network has 3,255 weakly connected components, including the 621 isolates. The largest connected component consists of 18,545 nodes. The network was filtered by times-cited $\geq$ 50 resulting in 2,644 nodes and 10,144 edges. The largest connected component of this network, shown in Fig 7 (middle), has 635 authors with 30 isolates removed. The figure uses the very same size- and color-coding as the co-author network for AI (Fig 7 top). The co-author labels are filtered by the number of times an author was cited ($\geq$ 700). Author 'Menon, Mani' has the largest number of citations (3,890) in this network. Menon is also the top-cited US author in the 30,784 co-authors network, along with seven other authors (marked in bold in Fig 7 middle table) that appear both in the largest connected component and listed as the top-cited US authors.

**Co-author U.S. map.** Fig 8 (middle left) shows the co-author network for robotics with nodes representing the number of citations and a darker hue indicating the first year of publication for a given author. The network shows a large concentration in the eastern and mid-U.S. states. Pittsburgh and Cambridge are the top-two cities, with robotics research being cited 6,486 and 5,672, respectively (see Fig 8, middle right).

**Topical evolution.** 'Electrical Engineering & Computer Science' has been the front-runner discipline in robotics for two decades both in publication and citation amount. It is also noticeable in Fig 9 (middle) that Health Related disciplines (e.g., 'Brain Research', 'Health Specialties') increased over time. Similar to AI, robotics show a steady increase within each of the 15 disciplines (see also S2 Fig in S1 Appendix). The top-five most cited papers, along with their publication year and total number of citations, are shown in S4 Table in S1 Appendix.

## Internet of things (IoT)

The term internet of things (IoT) refers to the network of physical devices such as phones, vehicles, or home appliances that have embedded electronics, software, sensors, actuators, and connectivity allowing them to collect, exchange, and act upon data. The dataset used here starts in 2006. Hence, there are no bursts or authors active before that year.

**WOS-top organizations and funding.** The top-10 funding organizations and the top-10 research organizations are given in Fig 6 (bottom right). As can be seen, funding from Chinese institutions is most often acknowledged, and three out of the top-10 research institutions are from China. NSF in the U.S. ranks second and two U.S. institutions are listed in the top-10 list. Funding by two European institutions is cited frequently, and four top-10 research institutions are from France.

**Burst of activity.** In the 348 NSF awards, there are 77 total bursts, with no term bursting more than once. The top-15 are shown in Fig 6 (bottom left). Similarly, for the 11,371 WOS publications there are 77 bursts total with no double bursts. There is no term that bursts in both sets. Burst weight is indicated by bar thickness, with "RFID" (Radio Frequency Identification) having the strongest burst of 62.2 in 2006-2013. It is important to point out that there is a clear separation of initial publications bursts, followed by several funding bursts, followed by a new set of publications bursts. The AI and robotics bursts were much more intermixed. Furthermore, there is a difference in terms of the strongest bursting weight between NSF and WOS. The most bursting term for WOS was "RFID" (62.16), whereas NSF had a much smaller value for its strongest burst, "Vehicles" (4.26).

**Key authors and collaboration networks.** The original IoT dataset has 23,691 nodes and 56,937 edges. In this network, there are 6,979 authors with more than three papers and 5,517 authors with more than five papers. The network has 3,345 weakly connected components, including the 506 isolates. The largest connected component consists of 11,438 nodes. After filtering by times-cited $\geq 5$, the network resulted in 6,939 nodes and 12,371 edges with the largest component of 585 with 98 isolates removed. The network was further filtered by times-cited $\geq 5$ with 585 authors plotted in Fig 7 (bottom). The figure uses the very same size- and color-coding as the co-author network for AI (Fig 7 top). The labels are filtered by the times-cited ($\geq 400$). The most cited author is 'Xu Ld' with 3,358 citations in the filtered network.

**Co-author U.S. map.** Fig 8 (bottom left) exhibits the co-author network for IoT. The lighter hue indicates more recent first publications by authors in this network. Kalamazoo and Norfolk are the top-two cities with IoT research being cited 902 and 635 respectively (see Fig 8 bottom right).

**Topical evolution.** The topical distribution of WOS publications on IoT is shown in Fig 9. Note that there are only seven papers published in 1998-2007. All other 11,364 journal papers were published in the recent decade. Most of the papers are in the 'Electrical Engineering & Computer Science' discipline with some in 'Chemical, Mechanical and Civil Engineering.' Note the larger number of papers in 'Electrical Engineering & Computer Science' published in venues such as *Wireless Personal Communications* (74 papers) and *International Journal of Distributed Sensor Network* (58) dealing with personal and complex implications of

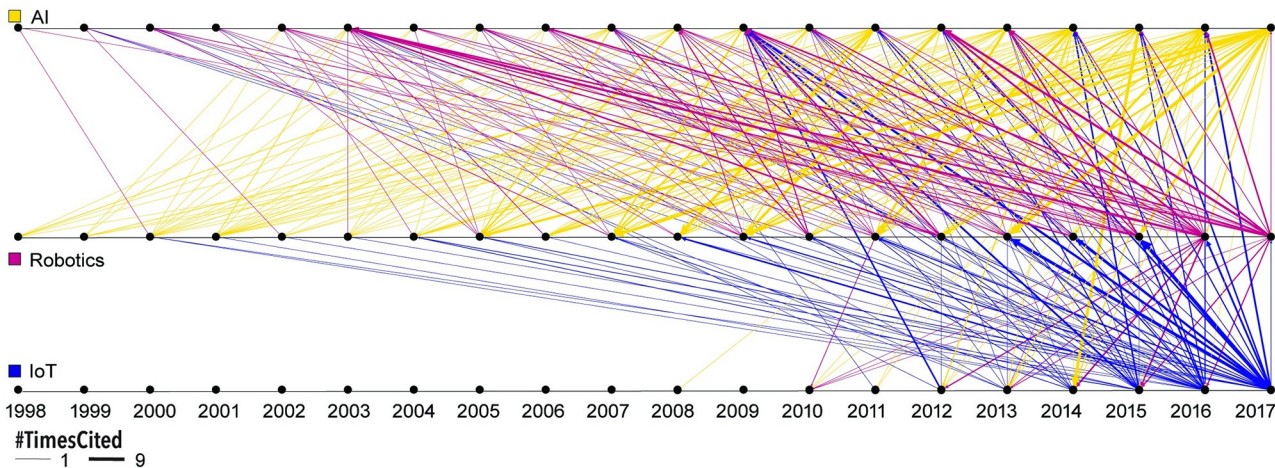

**Fig 10. Temporal convergence between AI (yellow), robotics (red), IoT (blue).**

IoT. It is also noticeable that IoT increases its topical coverage from three disciplines (1998-2007) to 13 disciplines (2007-2018), supporting evidence that IoT is a nascent field (see also S3 Fig in S1 Appendix). The top-five most cited papers plus publication year and total number of citations are shown in S4 Table in S1 Appendix.

## Convergence

Over the last 20 years, the three areas of "artificial intelligence," "IoT," and "robotics" have been merging. That is, there are more and more publications, funding awards, and keywords that are shared between pairs or even among all three of these areas. Fig 10 shows the increase in inter-citation linkages. Citations from papers in AI to papers in robotics and IoT are given in yellow; arrows are thickness-coded by the number of citations. Note that early arrows are rather thin while more recent citation links are thicker. As expected, only papers from earlier or the same year can be cited (i.e., arrows either point downwards or down-left). Citations from papers in robotics are given in red and many cite papers in AI.

Citations from papers in IoT are given in blue and they go back to AI and robotics papers as early as 2000; as the IoT dataset only covers papers published in 2004-2017, they only start citing papers from other domains in more recent years (2012-2017), particularly heavily citing robotics in 2013 and 2015 and AI in 2009. Of interest here, we see a spike in NSF awards for IoT research during those years (see Fig 1B).

## Validation: User studies

Decision makers from a variety of areas at NSWC Crane were invited to examine and help interpret initial versions of the visualizations related to AI in terms of readability, memorability, reproducibility, and utility. This study was approved by the Indiana University Institutional Review Board (protocol number 1809442778). This user study was administered as an online survey delivered via Qualtrics which took 30-50 minutes for participants to complete, see S1 Appendix for survey instrument. Five expert decision makers participated in the user study. Participants were presented with five visualizations on the topic of artificial intelligence. They were asked to complete tasks demonstrating their ability to interpret the visualizations and were asked to provide feedback on the utility of the visualizations in their particular line of work. Feedback collected during the user studies helped optimize algorithm and user interface

**Table 3. Summary from the expert qualitative opinions for three types of visualizations.**

| Topic | Top Organization | Top Agencies | Burst of Terms | Co-author Network | Topical Evolution |
|---|---|---|---|---|---|
| Strategic planning | ✓ | ✓ | ✓ | ✓ | ✓ |
| Potential partnership | ✓ | | | ✓ | |
| Research | | | ✓ | ✓ | ✓ |
| National importance | ✓ | ✓ | | | |
| Hiring | | | ✓ | ✓ | |

implementations and improve documentation of the results. Three visualizations are relevant for the work presented here.

The first visualization is a burst diagram, showing bursts of terms in funding awards and in publications. Burst diagrams can be particularly useful for understanding temporal relationships between funding and publication streams. Users speculated that burst rates may be tied to larger economic conditions, which affect funding streams and R&D investment. The visualization can also confirm strategic direction and identify subtle shifts in focus. For example, one user described how the "earlier burst was related more to neural networks, algorithms, and knowledge/expert systems. The recent burst seems to be related to large datasets, computer vision, and deep learning (more "big data" topics)." One can gain insight into areas that are receiving funding now, and may therefore see research advances in the future, which can help drive the development of proposals that will be relevant to granting organizations.

The second visualization showed top-10 subnetworks with the largest number of authorships. Users noted that the ability to understand which researchers are most prolific and which are working across disciplines was valuable. In the words of one user, "having an understanding of major authors in each area and how they interrelate allows me to determine who to work with in a given topic area and who may have a larger breadth of knowledge." The visualization could be used to identify potential collaborators as well as to deepen a general understanding of how researchers in this topic area are related.

The third visualization illustrated topical evolution visualized on a map of science. Users felt this map had the least direct application to their daily work. When asked to identify which visualizations were most relevant for their work, study participants identified both burst analysis and key authors and collaboration networks as highly relevant. This is consistent with results from the stakeholder needs analysis. Table 3 summarizes the feedback on the utility of each visualization for strategic planning, building potential partnerships, setting research agendas, determining national importance, and making hiring decisions.

## Discussion and outlook

This project used large-scale publication and funding data to support the analysis and visualization of key experts, institutions, publications, and funding in strategic areas of interest that were identified via a formal user needs analysis. Results of the study can be used to identify leading experts and potential investigators or reviewers; to detect emerging or declining areas; and to understand the role that funding agencies play in different countries and various topic areas.

Specifically, this paper used a formal stakeholder needs analysis to identify key insight needs together with strategic areas of interest. A detailed analysis of topic bursts in publication and funding data was performed; funding by international research organizations was compared; major authors in the U.S. were mapped geospatially; and the topical evolution was mapped for all three areas. Results were validated through a formal user study with

professionals working in these areas. Novel visualization algorithms such as the double-burst visualization in Fig 6 and the convergence visualization in Fig 10 led to actionable insights. The data, code, and workflows developed for and used in this paper have been documented and published on GitHub (https://github.com/cns-iu/AICoEvolution) so results can be reproduced and other topic areas can be studied.

The presented research has several limitations. First, the analysis uses two high-quality, high-coverage data sources (Web of Science and NSF Awards database) but other data (e.g., publication data from the arXiv preprint repository [25] or patents to capture technology evolution [7]) could be added in future studies to capture science and technology developments. Note that an inclusion of additional data sources would require disambiguation and cleaning of author names and geographical locations. While Web of Science provides information on paper citations, arXiv does not. Going forward, we are interested to explore additional datasets such as Federal Business Opportunities (FBO) data that would make it possible to gain a more comprehensive understanding of the S&T funding landscape. Secondly, we used a variety of metrics such as the number of publications and citations, publication sources (journals), disciplines and subdisciplines, authors and their co-authorship relations, authors' geographical locations, keywords, extracted terms (NLP MaxMatch feature engineering method), award $ amounts, type of organization and funding agency. Future work might also like to consider additional metrics such as authors' diversity proposed by [7] or topic diversity recently suggested by [25]. Finally, we focused on a small, static subset of keywords and their alternatives (IoT vs Internet of Things) found in abstracts, titles, or keywords and did not examine contextual or semantic changes of terms over time.

Experts that participated in the survey and user study expressed a strong interest in interactive data visualizations that would make it possible to zoom into specific subareas or to retrieve details on a specific author or paper or funding award. Going forward, we plan to develop interactive visual interfaces to near real-time datasets to support experts managing and evaluating research portfolios and making strategic decisions related to the systematic growth of different research areas.

## Supporting information

**S1 Appendix. This appendix contains S1-S6 Tables and S1-S3 Figs.**
(PDF)

## Acknowledgments

The authors would like to thank the anonymous reviewers for their expert comments. The authors also thank Perla Brown for design support, Tenzin Choeden and Jimmy Huang for excellent research assistance, and Todd Theriault for copy editing. This work uses Web of Science data by Clarivate Analytics provided by the Indiana University Network Science Institute. Thanks go to Joseph Brightbill from Clarivate Analytics Web of Science for providing WOS accession numbers.

## Author Contributions

**Conceptualization:** Katy Börner, Olga Scrivner, Lisel Record.

**Data curation:** Olga Scrivner, Michael Gallant, Haici Yang.

**Methodology:** Katy Börner, Olga Scrivner, Shutian Ma, Adam S. Martin, Haici Yang.

**Resources:** Jonathan M. Dilger.

**Software:** Michael Gallant, Shutian Ma.

**Validation:** Lisel Record.

**Visualization:** Katy Börner, Olga Scrivner, Leonard E. Cross, Haici Yang.

**Writing – original draft:** Katy Börner, Olga Scrivner.

**Writing – review & editing:** Jonathan M. Dilger.

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
