## [Decision Letter · Decision Letter 0]

11 Aug 2020

PONE-D-20-16726

Mapping the co-evolution of artificial intelligence, robotics, and the internet of things over 20 years (1998-2017)

PLOS ONE

Dear Dr. Scrivner,

Thank you for submitting your manuscript to PLOS ONE. After careful consideration, we feel that it has merit but does not fully meet PLOS ONE’s publication criteria as it currently stands. Therefore, we invite you to submit a revised version of the manuscript that addresses the points raised during the review process.

First, I'd like to commend your team of collaborators for considering such a complex meta-study. I concur with both reviewers that, as is, the manuscript is not technically sound, which is a critical issue for a journal like PLoS ONE. I therefore ask you to thoroughly revise their manuscript to address the flaws in methodology, as well as all the other detailed comments from the reviewers.

We look forward to receiving your revised manuscript.

Kind regards,

Roland Bouffanais, Ph.D.

Academic Editor

PLOS ONE

Journal Requirements:

2. We note that [Figure(s) #] in your submission contain [map/satellite] images which may be copyrighted. All PLOS content is published under the Creative Commons Attribution License (CC BY 4.0), which means that the manuscript, images, and Supporting Information files will be freely available online, and any third party is permitted to access, download, copy, distribute, and use these materials in any way, even commercially, with proper attribution. For these reasons, we cannot publish previously copyrighted maps or satellite images created using proprietary data, such as Google software (Google Maps, Street View, and Earth). For more information, see our copyright guidelines: http://journals.plos.org/plosone/s/licenses-and-copyright.

1.    You may seek permission from the original copyright holder of Figure(s) [#] to publish the content specifically under the CC BY 4.0 license. 

Additional Editor Comments (if provided):

First, I'd like to commend the authors for considering such a complex meta-study. I concur with both reviewers that, as is, the manuscript is not technically sound, which is a critical issue for a journal like PLoS ONE. I therefore ask the authors to thoroughly revise their manuscript to address the flaws in methodology, as well as all the other detailed comments from the reviewers.

Reviewers' comments:

Reviewer's Responses to Questions

**Comments to the Author**

1. Is the manuscript technically sound, and do the data support the conclusions?

Reviewer #1: Partly

Reviewer #2: Partly

2. Has the statistical analysis been performed appropriately and rigorously? 

Reviewer #1: N/A

Reviewer #2: Yes

3. Have the authors made all data underlying the findings in their manuscript fully available?

Reviewer #1: No

Reviewer #2: Yes

4. Is the manuscript presented in an intelligible fashion and written in standard English?

Reviewer #1: Yes

Reviewer #2: Yes

5. Review Comments to the Author

Reviewer #1: The article analyzes three science and technology areas, namely AI, robotics, and IoT, and their coevolution over twenty years. Using data on funding, publications, and citations, the study produces visualizations of burst topics, co-author networks, and inter-citations over time, etc. A user study was conducted to examine the usefulness and utility of related visualizations.

On page 3, Table 1, 0 NSF awards for "human systems integration"? Is this related to HCI? Is it a coding error?

On page 6/7, is figure 2 a hypothetical example or based on real data? What are the parameters of the burst analysis? How were the exact keywords (both single words and phrases) identified? It is interesting to see Support Vector Machines had a 10-year burst (2008 - 2018) when other innovations such as deep learning should have gained more popularity in the 2010s.

On page 9/10, the top authors and co-author component analyses are useful, though it is difficult to read and interpret co-author networks (figure 7) and those with geographical overlay (figure 8).

Figure 9 is very interesting. However, it is again visually challenging to compare the two decade periods of 1998-2007 vs. 2008-2017. Perhaps there is a way to overlay the two maps or create another map to contrast the differences (changes/growth) between the two decades.

Figure 10 is useful in depicting the coevolution and inter-citations of the three fields. It would be better (and simpler) to run the overall statistics and show the total number of inter-citations over time.

How many subjects participated in the user study? I cannot find the number in the manuscript. Also, the user study analysis is anecdotal and lacks details about the structure and coding of related questions.

Overall, the paper is exploratory and provides interesting results about the development of three important research areas in science and technology. However, it lacks significant results and major findings as a research article.

Reviewer #2: The authors aim to demonstrate the emergence and convergence of three fields of research: Artificial intelligence (AI), robotics and Internet of things (IoT), using publication, funding and scholarly network metrics. The authors also aim to show how the novel visualizations and concepts introduced in this work could better inform directions for interdisciplinary research or new research areas.

While the article is generally well written, and is clear with respect to its intended contributions, I think it could benefit from a lot more clarity in terms of the methods considered, the presentation, the interpretation of the results and how they tie back to the primary claims of the paper. That is, how exactly do the results create a tangible potential for interdisciplinary research? In addition, the current results consider AI, robotics and IoT (fields that are known to be inter-related) and establishes their inter-dependencies over the years through various metrics/visualizations. However, it would have been more interesting to examine randomly selected fields, and determine retrospectively, whether there exists a potential for interdisciplinary research in the future. I also suggest restructuring certain portions of the article to enhance clarity. The quality and presentation of all figures must be improved. I have provide my detailed comments below:

1. line 12: Please define emergence and convergence, and clearly state which one (or both) is the focus of the paper

2. line 23: Is this statement based on your data or some previous study? I could also imagine new authors entering a research area and word burst occurring without it being very interdisciplinary.

3. lines 38-43: It is better to clearly state in a single paragraph the novelty and contributions of your work. What is it that separates it from previous work?

4. lines 58-63: Section numbers are mentioned, but sections are not numbered. Same issue throughout the article.

5. line 78: Was there any basis for selecting those 8 topics?

6. Wouldn’t it be better to analyze all 8 fields and show how your visualizations/metrics inform directions for future research?

7. Line 83-85: Why use publication and funding data to select from the list of fields? Doesn’t it make more sense to select fields that show the most growth most recently? For example, if a field X has a very high number of total funding awards and publications, it need not necessarily mean that the field is currently relevant. Most of the funding may have been obtained say, before 2005. So if one must select emerging areas, it must be based on the current trend of funding/publications and not the total number.

8. Lines 98-115: It is not clear whether these visualizations are the ones that you have shown in the figures of this paper. I think they are not. If that is the case, why not include sample of the visualizations that you refer to in line 99?

9. Line 137- why capitals?

10. To me it is not clear how these query terms would work. For example, there are probably several AI papers that were written in 1998 which would not contain the words ‘Artificial Intelligence’ or ‘Deep learning’. How would these papers be classified? Would they be ignored for the purposes of this paper? If so, it should be clearly mentioned.

11. In Fig 1, do the colored curves correspond to citations or publications?. Figure looks grainy. Quality should be enhanced.

12. Line 148: Should be 2004, not 2014.

13. In fig 1, why is the dotted line (total citations) showing a downward trend, while none of the individual fields (colored curves) seem to be decreasing? Again, are colored curves citations or publications?

14. Line 178: “There was no award for a project in all three focus areas.” - Change to “There was no award for a project involving all three focus areas.”

15. Fig2: Why not also show ‘burstiness’ through line thickness or by using colors?

16. Fig2 : Why plot both funding and publication bursts in the same plot? Wouldnt it be clearer to have separate plots for each?

17. Fig 5: What determines the position of sub-disciplines?

18. Fig 5 caption should be more descriptive.

19. Fig 3: Publication year shows 1970-1970

20. Fig3: Does the edge color represent the year of the latest co-authored papers?

21. Fig 4: In the bottom table of the figure, papers are referred to as A1,A2, which is inconsistent with the notation for papers in Fig 3 (P1,P2 etc.,)

22. Fig 5: The text written in yellow is impossible to read. In general, the figure is too grainy.

23. Fig 5: Why aren’t all colors mentioned in the legend?

24. Results section: Introduction to the fields of AI, robotics and IoT can be moved to after tge stakeholder analysis. There is no point introducing these topics in page 8, when it has been continuously been mentioned/discussed from pages 1-8. Same goes for the sub-section WOS-top organizations and funding. Better to talk about this immediately after the stakeholder analysis.

25. Fig 6: Thickness is used to depict ‘burstiness’, but this was not mentioned in the ‘Burst detection and visualization’ section

26. Lines 383-481: Why not reorganize this to talk about “Burst of activity” for all three fields, then move to “Key authors and collaboration networks” for all three and so on? This way, it would be easier to compare and show the correlations and contrasts between different fields.

27. Line 488: I did not understand the sentence about the arrow thickness. Do you mean arrows corresponding to earlier works are thinner? What if the arrow connects an early work to one of the latest works? Also, there is no perceivable difference in thickness in the figure.

28. Lines 528-535: Questions asked to users could be included in the appendix

29. Line 543: bursts

30. Clarity of all figures need to be improved

31. Fig 9: ‘Brain research’ is missing in the legend.

32. Fig 9: It would be much easier to see the differences between the sub-figures if the names of the fields are removed from the figure. They are color coded with the legend anyway, so you can do this.

33. Figure 10: Did not understand the significance of the legend ‘1 and 9’

34. Line 381: Cannot see the steady increase in the figure.

35. In the discussion, also clearly mention the drawbacks and challenges of your proposed visualizations and approach. What would be the challenges in scaling this up, say, if one wanted to examine 100 fields instead of 3? How would the inclusion of international collaborations affect your approach? What would need to be considered while taking into account currency differences when it comes to funding on a global scale?

6. PLOS authors have the option to publish the peer review history of their article (what does this mean?). If published, this will include your full peer review and any attached files.

Reviewer #1: No

Reviewer #2: **Yes: **Thommen George Karimpanal

---

## [Author Response · Author response to Decision Letter 0]

28 Sep 2020

Dear Editor:

We are grateful for the extremely helpful review of our manuscript on Mapping the co-evolution of artificial intelligence, robotics, and the internet of things over 20 years (1998-2017).

We submit here a revised version of our article, which incorporates the changes recommended by the reviewers. We also include a point-by-point response to each of their comments.

Thank you again for your continued consideration of our work. We hope this revision is sufficiently responsive to the first round of feedback to merit publication, but we stand ready to make whatever adjustments you deem necessary.

In response to comments from Reviewer 1:

1. On page 3, Table 1, 0 NSF awards for "human systems integration"? Is this related to HCI? Is it a coding error?

Response

Thank you for raising this question. The NSF award portal was utilized to query for terms suggested by the stakeholders. “Human systems integration” was one of the suggested terms and the result of the query was zero award with selected filters applied.

2. On page 6/7, is figure 2 a hypothetical example or based on real data? What are the parameters of the burst analysis? How were the exact keywords (both single words and phrases) identified? It is interesting to see Support Vector Machines had a 10-year burst (2008 - 2018) when other innovations such as deep learning should have gained more popularity in the 2010s.

Response

Thank you for helping to clarify this. We added the following text to line 228: “Fig 2 explains this new burst visualization using a hypothetical example of six compound terms (T) that burst between 2004-2018." The algorithm used to extract keywords is described in section Keyword Extraction via MaxMatch (page 6). The parameters for burst analysis are explained in section Burst detection and visualization (page 7).

3. On page 9/10, the top authors and co-author component analyses are useful, though it is difficult to read and interpret co-author networks (figure 7) and those with geographical overlay (figure 8).

Response

Thank you for raising this concern. Our aim is to display the overall structure of the co-author network by showing major nodes and connections; network clusters, backbones, and density; and author impact via node size (#citations) and co-author network evolution (the darker a node, the older the authorship). In the Discussion and Outlook section, we discuss the need for interactive visualizations where hovering over an author would reveal his/her name, #papers, #citations, among others.

4. Figure 9 is very interesting. However, it is again visually challenging to compare the two decade periods of 1998-2007 vs. 2008-2017. Perhaps there is a way to overlay the two maps or create another map to contrast the differences (changes/growth) between the two decades.

Response

Thank you for suggesting this. The values for each period are provided in Supplemental material.

We also added to line 382 (p.14): (see also S1 Fig and S1 Table in S1 Appendix).

5. Figure 10 is useful in depicting the coevolution and inter-citations of the three fields. It would be better (and simpler) to run the overall statistics and show the total number of inter-citations over time.

Response

Thank you for suggesting this. We now added a convergence summary table (excel file) to our GitHub repository which is available at https://github.com/cns-iu/AICoEvolution/.

6. How many subjects participated in the user study? I cannot find the number in the manuscript. Also, the user study analysis is anecdotal and lacks details about the structure and coding of related questions.

Response

Thank you for this question. In the revised version, we added the number of subjects in the user study (line 503, p.18). Demographic information and survey instruments are provided in S1 Appendix and our GitHub repository available at https://github.com/cns-iu/AICoEvolution/.

7. Overall, the paper is exploratory and provides interesting results about the development of three important research areas in science and technology. However, it lacks significant results and major findings as a research article.

Response

With due respect, we would like to argue that the paper presents a rather comprehensive analysis and comparison of three strategically important research areas that are co-evolving. It presents a novel visualization of co-bursts and a novel visualization of convergence. As confirmed by the formal user study, the study results and the novel visualizations are informative and actionable by experts that need to prioritize investments and make strategic decisions in these areas.

In response to comments from Reviewer 2:

1. While the article is generally well written, and is clear with respect to its intended contributions, I think it could benefit from a lot more clarity in terms of the methods considered, the presentation, the interpretation of the results and how they tie back to the primary claims of the paper. That is, how exactly do the results create a tangible potential for interdisciplinary research? In addition, the current results consider AI, robotics and IoT (fields that are known to be inter-related) and establishes their inter-dependencies over the years through various metrics/visualizations. However, it would have been more interesting to examine randomly selected fields, and determine retrospectively, whether there exists a potential for interdisciplinary research in the future. I also suggest restructuring certain portions of the article to enhance clarity. The quality and presentation of all figures must be improved. I have provide my detailed comments below:

Response

The original submission includes high resolution vector versions of all figures. Please kindly use those when reviewing the paper.

Using 3 topics of interest as an illustration, we identified leading international academic organizations, funding agencies, top researchers relevant to each topic as well as convergence between topics enabling new pathways for a stakeholder to develop future investment and funding opportunities. We revised the abstract to make this clear.

We agree that this is an alternative approach to conduct valuable research. However, the work presented here is more pragmatic--it starts with a set of research areas that are of strategic interest to different governmental labs as well as industry representatives so that study results can directly support data-driven decision making.

2. line 12: Please define emergence and convergence, and clearly state which one (or both) is the focus of the paper

Response

The paper focuses on emergence using a 4-attribute model: novelty, persistence, growth and research community (line 15, p.1) and convergence of three emerging areas. Emerging areas (topics) were identified based on publication and funding data (line 35-42, p.2).

3. line 23: Is this statement based on your data or some previous study? I could also imagine new authors entering a research area and word burst occurring without it being very interdisciplinary.

Response

Yes, relevant prior work is cited in line 17: (Gao et al., 2011). 

4. lines 38-43: It is better to clearly state in a single paragraph the novelty and contributions of your work. What is it that separates it from previous work?

Response

Our research presents novel visualizations to examine the emergence, growth and convergence of scientific research areas. 

5. lines 58-63: Section numbers are mentioned, but sections are not numbered. Same issue throughout the article.

Response

Thank you. We changed section numbers to section names as suggested.

6. line 78: Was there any basis for selecting those 8 topics?

Response

A user needs analysis (see Section Stakeholder needs analysis) was conducted to select these 8 topics.

7. Wouldn’t it be better to analyze all 8 fields and show how your visualizations/metrics inform directions for future research?

Response

The top-three, strategically most valuable fields were selected to fit the comparison in a Plos One paper. All workflows are well documented and code is available on GitHub and they can be re-run for the remaining five or any other field.

8. Line 83-85: Why use publication and funding data to select from the list of fields? Doesn’t it make more sense to select fields that show the most growth most recently? For example, if a field X has a very high number of total funding awards and publications, it need not necessarily mean that the field is currently relevant. Most of the funding may have been obtained say, before 2005. So if one must select emerging areas, it must be based on the current trend of funding/publications and not the total number.

Response

Thank you for your suggestion. We selected fields that are of strategic interest to different governmental labs as well as industry representatives so that study results can directly support data-driven decision making. From the list of strategic areas, we selected the top three based on their NSF awards and publications.

9. Lines 98-115: It is not clear whether these visualizations are the ones that you have shown in the figures of this paper. I think they are not. If that is the case, why not include sample of the visualizations that you refer to in line 99?

Response

We included these visualizations in the GitHub repository with questions used for user needs and user studies. We used an Artificial Intelligence topic for user studies and we made some adjustments to figures as they were distributed as a paper-based questionnaire.

10. Line 137- why capitals?

Response

These are the examples of unprocessed keyword terms. We replaced them by lower character strings.

11. To me it is not clear how these query terms would work. For example, there are probably several AI papers that were written in 1998 which would not contain the words ‘Artificial Intelligence’ or ‘Deep learning’. How would these papers be classified? Would they be ignored for the purposes of this paper? If so, it should be clearly mentioned.

Response

Excellent point. As a field evolves, there are changes in terminology. In addition, not all authors would adhere to providing standard (widely accepted) terms. However, in dealing with a large scale dataset, we focused only on 3 words and their alternatives (IoT vs Internet of Things) found in abstracts, titles, or keywords. The extracted publications served as a baseline dataset that was used to collect pertinent keywords. That is, any term that at that time was deemed important to include as a keyword becomes a part of a compound term set.

12. In Fig 1, do the colored curves correspond to citations or publications?. Figure looks grainy. Quality should be enhanced.

Response

The original submission includes high resolution vector versions of all figures. Please kindly use those when reviewing the paper. 

The “colored curves” correspond to the number of publications per each individual domain. This information is also available via the Figure 1A description which reads: “Yellow solid line represents AI publications, red solid line - robotics publications, and blue solid line - IoT publications.” 

Text description for Figure 1B: ” Yellow solid line represents AI funding, red solid line - robotics funding, and blue solid line - IoT funding. 

We added this information to the main paper text as well.

13. Line 148: Should be 2004, not 2014.

Response

Thank you for pointing this out. The text has been changed to “between 2007-2011” we observe a nascent field of IoT (blue solid line) showing a sharp increase.

14. In fig 1, why is the dotted line (total citations) showing a downward trend, while none of the individual fields (colored curves) seem to be decreasing? Again, are colored curves citations or publications?

Response

Excellent question. We added the following text to the main text and the figure caption: “The dotted line in Figure 1A represents the total number of citations for all three domains. Note that papers published in recent years did not yet have time to acquire a high citation count.”

15. Line 178: “There was no award for a project in all three focus areas.” - Change to “There was no award for a project involving all three focus areas.”

Response

This was changed. Thank you for suggesting.

16. Fig2: Why not also show ‘burstiness’ through line thickness or by using colors?

Response

Thank you for your suggestion. Burstiness (strength) is already encoded in the bar height (thickness). The hypothetical example in Figure 2 uses the equal bar height for each term, as described in line 229. To clarify this, we added the following text “the height (thickness) shows a burst strength.”

17. Fig2 : Why plot both funding and publication bursts in the same plot? Wouldnt it be clearer to have separate plots for each?

Response

Two plots would make each easier to examine. However, we are interested to compare bursts across datasets. This novel visualization makes it possible to do just this--the start and end dates but also burst strength can be compared. In addition, the co-occurrence of bursting terms can be easily discovered. We added this explanation to the main text.

18. Fig 5: What determines the position of sub-disciplines?

Response

The position of sub-disciplines is prescribed by the UCSD Map of Science classification system, please see details in [1]. The reference is also provided in the text.

19. Fig 5 caption should be more descriptive.

Response

Thank you for your suggestion. We added the following text to the caption: "A process of mapping scientific journal names to discipline and (sub)discipline topic (left) and the projection of journal topics into 2D spatial position (right)."

20. Fig 3: Publication year shows 1970-1970

Response

Thank you for pointing this out. We changed the legend to First Year Last Year.

21. Fig3: Does the edge color represent the year of the latest co-authored papers?

Response

In this simple network performed via Sci2, edges denote co-authorship relations and are color coded by year of the first joint publication and thickness coded by the number of joint papers.

22. Fig 4: In the bottom table of the figure, papers are referred to as A1,A2, which is inconsistent with the notation for papers in Fig 3 (P1,P2 etc.,)

Response

Thank you for this suggestion. We updated the table in Fig 4.

23. Fig 5: The text written in yellow is impossible to read. In general, the figure is too grainy.

Response

The original submission includes high resolution vector versions of all figures. Please kindly use those when reviewing the paper. We also added a shadow to the yellow labels.

24. Fig 5: Why aren’t all colors mentioned in the legend?

Response

This figure in the Method section explains the general process of computing a science map from publication data using hypothetical data.

The map has 13 disciplines that are color coded and labelled in the same color. We added explanatory text to the main paper to make this easier to understand.

25. Results section: Introduction to the fields of AI, robotics and IoT can be moved to after tge stakeholder analysis. There is no point introducing these topics in page 8, when it has been continuously been mentioned/discussed from pages 1-8. Same goes for the sub-section WOS-top organizations and funding. Better to talk about this immediately after the stakeholder analysis.

Response

The very brief introductions to specifics of the three fields serve as an introduction to each of the three research areas. We now explain this structure and intent in the very beginning of the Results section.

26. Fig 6: Thickness is used to depict ‘burstiness’, but this was not mentioned in the ‘Burst detection and visualization’ section

Response

Thank you for your suggestion. To clarify this, we added the following text “the height (thickness) shows a burst strength.” in Burst detection and visualization section.

27. Lines 383-481: Why not reorganize this to talk about “Burst of activity” for all three fields, then move to “Key authors and collaboration networks” for all three and so on? This way, it would be easier to compare and show the correlations and contrasts between different fields.

Response

We thank the reviewer for this suggestion. We presented topics separately to provide a more holistic picture of topic evolution and then merged them to depict their convergence.

28. Line 488: I did not understand the sentence about the arrow thickness. Do you mean arrows corresponding to earlier works are thinner? What if the arrow connects an early work to one of the latest works? Also, there is no perceivable difference in thickness in the figure.

Response

The thickness of arrows corresponds to the number of citations (line 488). The visualization shows that the early papers do not have many converging citations (across disciplines), that is they are thinner. The legend shows the thickness on the scale between 1 and 9. We added a legend title “#TimesCited”.

29. Lines 528-535: Questions asked to users could be included in the appendix

Response

User needs and study questions are included in the GitHub repository created for this paper.

30. Line 543: bursts

Response

This was changed. Thank you for suggesting.

31. Clarity of all figures need to be improved

Response

The original submission includes high resolution vector versions of all figures. Please kindly use those when reviewing the paper.

32. Fig 9: ‘Brain research’ is missing in the legend.

Response

Thank you for pointing this out. We updated the legend in Fig 9.

33. Fig 9: It would be much easier to see the differences between the sub-figures if the names of the fields are removed from the figure. They are color coded with the legend anyway, so you can do this.

Response

The original submission includes high resolution vector versions of all figures. Please kindly use those when reviewing the paper.

34. Figure 10: Did not understand the significance of the legend ‘1 and 9’

Response

The thickness of arrows corresponds to the number of citations (line 488). The legend shows the thickness on the scale between 1 and 9. We added a legend title “#TimesCited”.

35. Line 381: Cannot see the steady increase in the figure.

Response

We made the following modification to line 381-382: “The topical coverage for AI has increased for all scientific disciplines with the highest publication change in Electrical Engineering & Computer Science (10,391) and Chemical, Mechanical & Civil Engineering (6,283), and Social Sciences (2680) (see also S1 Fig and S1 Table in S1 Appendix)”

36. In the discussion, also clearly mention the drawbacks and challenges of your proposed visualizations and approach. What would be the challenges in scaling this up, say, if one wanted to examine 100 fields instead of 3? How would the inclusion of international collaborations affect your approach? What would need to be considered while taking into account currency differences when it comes to funding on a global scale?

Response

We thank the reviewer for these questions. First, Gephi is suitable for large-scale network visualizations (see Yifan Hu and OpenOrd plugins) and widely used with large biomedical data (e.g., genes, proteins) [2].

Secondly, the Web of Science is a global repository of scientific papers. Our top funding agencies and organizations analyses show who is leading in a particular topic of interest. For instance, AI research leading institutions are global: U.S., India, China. Similarly, the co-authors network combines both national and international connections. US map was selected as a geographic layout in this analysis, however, the international map is available as long as there are authors’ coordinates in the metadata.

Finally, including international funding will require an additional investigation whether such data is available, and then converting funding via currency exchange. 

References

1. Börner K, Klavans R, Patek M, Zoss AM, Biberstine JR, Light RP, et al. Designand update of a classification system: The UCSD map of science. PLoS ONE.2012;7(7):e39464. doi:10.1371/journal.pone.0039464.

2. Pavlopoulos, G. A., Paez-Espino, D., Kyrpides, N. C., & Iliopoulos, I. (2017). Empirical Comparison of Visualization Tools for Larger-Scale Network Analysis. Advances in Bioinformatics. Hindawi Limited. https://doi.org/10.1155/2017/1278932

---

## [Decision Letter · Decision Letter 1]

16 Oct 2020

PONE-D-20-16726R1

Mapping the co-evolution of artificial intelligence, robotics, and the internet of things over 20 years (1998-2017)

PLOS ONE

Dear Dr. Scrivner,

Thank you for submitting your manuscript to PLOS ONE. After careful consideration, we feel that it has merit but does not fully meet PLOS ONE’s publication criteria as it currently stands. Therefore, we invite you to submit a revised version of the manuscript that addresses the points raised during the review process.

We look forward to receiving your revised manuscript.

Kind regards,

Roland Bouffanais, Ph.D.

Academic Editor

PLOS ONE

Additional Editor Comments (if provided):

Reviewer #2 still has some issues that should probably be addressed in a second round of revision.

Reviewers' comments:

Reviewer's Responses to Questions

**Comments to the Author**

1. If the authors have adequately addressed your comments raised in a previous round of review and you feel that this manuscript is now acceptable for publication, you may indicate that here to bypass the “Comments to the Author” section, enter your conflict of interest statement in the “Confidential to Editor” section, and submit your "Accept" recommendation.

Reviewer #1: All comments have been addressed

Reviewer #2: (No Response)

2. Is the manuscript technically sound, and do the data support the conclusions?

Reviewer #1: Partly

Reviewer #2: Partly

3. Has the statistical analysis been performed appropriately and rigorously? 

Reviewer #1: No

Reviewer #2: N/A

4. Have the authors made all data underlying the findings in their manuscript fully available?

Reviewer #1: Yes

Reviewer #2: Yes

5. Is the manuscript presented in an intelligible fashion and written in standard English?

Reviewer #1: Yes

Reviewer #2: Yes

6. Review Comments to the Author

Reviewer #1: (No Response)

Reviewer #2: 1. While I agree with the authors’ argument that the work examines areas of strategic interest, my concern is that these selected areas happen to be very correlated with each other. This raises the question of how useful the presented approach would be when the selected areas are not correlated with each other. For example, if a policy maker wanted to make inferences about say, ‘Power and energy management’ and ‘robotics’ (or any two fields that don’t seem to be immediately/obviously correlated), I wonder how the approach would fare.

My point is that currently, the authors show how their approach can be useful when related fields are considered. While this is useful, I think it is also equally important to show what the results would look like when your approach is used to evaluate seemingly unrelated fields of research. I believe replicating at least some of the experiments on other non-correlated fields would serve to strengthen the contribution of this paper, which is also a concern raised by Reviewer 1. Otherwise, it would seem like the authors only considered the convenient choice of correlated fields of research.

2. The authors’ response to point 8. from the first round of reviews did not really address my concern. I am not suggesting the analysis be re-done based on the current trend (which could be quantified by say, the rate of change in the citations and publications), but I think the authors need to justify why using the total citations and publications is more relevant than using the current trends of these quantities, when it comes to the strategic interests of governments and industries. At the very least, the authors should acknowledge that other measures (other than the total pubs.+cites.) could be also be used.

3. The point about changes in terminology is an important one (comment 11. from the first round of reviews). I hope the authors include the points mentioned in their response, to the main manuscript.

4. The text overlaid on top of the graph in Fig 9 can be removed (point 33. from 1st round of reviews). This could make the figure significantly less cluttered, without any loss of information, as the corresponding color codes are already mentioned in the legend.

5. Regarding point 36., I still think it would be valuable to add a section on the limitations and future issues that remain to be addressed. This could be useful for those aiming to develop similar visualization tools in the future.

7. PLOS authors have the option to publish the peer review history of their article (what does this mean?). If published, this will include your full peer review and any attached files.

Reviewer #1: No

Reviewer #2: **Yes: **Thommen Karimpanal George

---

## [Author Response · Author response to Decision Letter 1]

11 Nov 2020

Dear Editor:

Thank you for the detailed review of our manuscript on Mapping the co-evolution of artificial intelligence, robotics, and the internet of things over 20 years (1998-2017).

We submit here a revised version of our paper, which implements many of the suggestions by Reviewer 2. We also include a point-by-point response to each of the comments.

Thank you again for your continued consideration of our work. We hope this revision addresses all remaining concerns and merits publication, but we stand ready to make whatever adjustments you deem necessary.

In response to comments from Reviewer 2:

1. While I agree with the authors’ argument that the work examines areas of strategic interest, my concern is that these selected areas happen to be very correlated with each other. This raises the question of how useful the presented approach would be when the selected areas are not correlated with each other. For example, if a policy maker wanted to make inferences about say, ‘Power and energy management’ and ‘robotics’ (or any two fields that don’t seem to be immediately/obviously correlated), I wonder how the approach would fare. My point is that currently, the authors show how their approach can be useful when related fields are considered. While this is useful, I think it is also equally important to show what the results would look like when your approach is used to evaluate seemingly unrelated fields of research. 

Response

Thank you for sharing your concern. Many strategic decision makers (research team leads, funders, teachers) are very interested to understand the interplay of research areas that co-evolve and impact each others’ growth). The paper presents general workflows and an exemplary analysis of three research areas that have high R&D impact and value.

Please do note that the same workflows can be used to understand the evolution of unrelated research areas. For example, data on ‘Power and energy management’ and ‘robotics’ can be compiled via the Web of Science and NSF award portals using relevant query terms. The NLP MaxMatch algorithm for keywords extraction and other workflows on Github https://github.com/cns-iu/AICoEvolution/ can be used to analyze and visualize these research areas. Even if the research areas are unrelated and no matching terms are found (aka null convergence), emerging topical areas in published work and awards, bursts, and co-author networks can be studied. However, this is not the main focus of the present paper.

2. I believe replicating at least some of the experiments on other non-correlated fields would serve to strengthen the contribution of this paper, which is also a concern raised by Reviewer 1. Otherwise, it would seem like the authors only considered the convenient choice of correlated fields of research.

Response

Just to reiterate, the paper introduces two novel visualizations (co-bursts and convergence) that are particularly valuable for studying the convergence of research areas. Both visualizations were exemplarily used to map three research areas of strategic importance and both were evaluated by domain experts. 

As we write in our original response to Reviewer 2, “the work presented here is more pragmatic--it starts with a set of research areas that are of strategic interest to different governmental labs as well as industry representatives so that study results can directly support data-driven decision making.”

Reviewer 1 was originally concerned about the “lack of significant results and major findings”) and not the choice of the research areas. In our recent revision, we pointed out that the paper introduces two novel visualizations (co-bursts and convergence) that were evaluated by domain experts which fully addressed the concerns of Reviewer 1. 

3. The authors’ response to point 8. from the first round of reviews did not really address my concern. I am not suggesting the analysis be re-done based on the current trend (which could be quantified by say, the rate of change in the citations and publications), but I think the authors need to justify why using the total citations and publications is more relevant than using the current trends of these quantities, when it comes to the strategic interests of governments and industries. At the very least, the authors should acknowledge that other measures (other than the total pubs.+cites.) could be also be used.

Response

Thank you for clarifying your concern. In our approach, we used a variety of metrics in addition to #publication and #citations, such as publication source (journals), discipline and subdiscipline, authors and their co-authorship relations, authors’ geographical locations, keywords, extracted terms (NLP MaxMatch feature engineering method), award $ amount, type of organization and funding agency.

We added the following text (p.2): “Recent studies introduced several novel NLP methods to measure research diversity and interdisciplinarity. For example, topic modeling has been used to measure the degree of topic diversity (Yegros-Yegros et al., 2015), Shannon’s entropy measure was applied to compute technological diversity using EU-funded nanotechnology projects data (Paez-Aviles et al., 2018). 

4. The point about changes in terminology is an important one (comment 11. from the first round of reviews). I hope the authors include the points mentioned in their response, to the main manuscript.

Response

Thank you for suggesting this. We added to the study limitation (p.15): “Finally, we focused only on a small subset of keywords and their alternatives (IoT vs Internet of Things) found in abstracts, titles, or keywords and did not examine contextual or semantic changes of terms over time.”

5. The text overlaid on top of the graph in Fig 9 can be removed (point 33. from 1st round of reviews). This could make the figure significantly less cluttered, without any loss of information, as the corresponding color codes are already mentioned in the legend.

Response

The visualization was computed using the Make-A-Vis tool. We prefer to keep the current figure to support easy and complete reproducibility of the presented workflow. This is analogous to a map of the world which would also show names of major geographic areas, e.g., continents or countries.

6. Regarding point 36., I still think it would be valuable to add a section on the limitations and future issues that remain to be addressed. This could be useful for those aiming to develop similar visualization tools in the future.

Response

Thank you for this suggestion. We added the following text to (p.15): “The presented research has several limitations. First, our analysis is based on two data sources: Web of Science and NSF Awards projects. Future studies might like to include publication data from the ArXiv preprint repository (Klinger et al., 2020) or patents to capture technology evolution (Paez-Aviles et al., 2018). Secondly, we used a variety of metrics such as the number of publication and citations, publication source (journals), discipline and subdiscipline, authors and their co-authorship relations, authors’ geographical locations, keywords, extracted terms (NLP MaxMatch feature engineering method), award $ amount, type of organization and funding agency. Future work might also like to consider authors’ diversity as proposed by (Paez-Aviles et al., 2018) and topic diversity (Weitzmn and Rao-Stirling) proposed by (Klinger et al., 2020). Finally, we focused on a small, static subset of keywords and their alternatives (IoT vs Internet of Things) found in abstracts, titles, or keywords and did not examine contextual or semantic changes of terms over time.” 

References

Klinger, J., Mateos-Garcia, J., & Stathoulopoulos, K. (2020). A narrowing of AI research. ArXiv.

Páez-Avilés, C., Van Rijnsoever, F. J., Juanola-Feliu, E., & Samitier, J. (2018). Multi-disciplinarity breeds diversity: the influence of innovation project characteristics on diversity creation in nanotechnology. Journal of Technology Transfer, 43(2), 458–481. https://doi.org/10.1007/s10961-016-9553-9

---

## [Editor Report · Decision Letter 2]

13 Nov 2020

Mapping the co-evolution of artificial intelligence, robotics, and the internet of things over 20 years (1998-2017)

PONE-D-20-16726R2

Dear Dr. Scrivner,

We’re pleased to inform you that your manuscript has been judged scientifically suitable for publication and will be formally accepted for publication once it meets all outstanding technical requirements.

Kind regards,

Roland Bouffanais, Ph.D.

Academic Editor

PLOS ONE
---

## [Editor Report · Acceptance letter]

18 Nov 2020

PONE-D-20-16726R2 

Mapping the co-evolution of artificial intelligence, robotics, and the internet of things over 20 years (1998-2017) 

Dear Dr. Scrivner:

I'm pleased to inform you that your manuscript has been deemed suitable for publication in PLOS ONE. Congratulations! Your manuscript is now with our production department. 

Kind regards, 

on behalf of

Professor Roland Bouffanais 

Academic Editor

PLOS ONE